

# A fully consistent and conservative vertically adaptive coordinate system for SLIM 3D v0.4, a DG finite element hydrodynamic model, with an application to the thermocline oscillations of Lake Tanganyika

Philippe Delandmeter[1], Jonathan Lambrechts[1], Vincent Legat[1], Valentin Vallaeys[1], Jaya Naithani[1], Wim Thiery[2,3], Jean-François Remacle[1], and Eric Deleersnijder[4,5]

[1]Université catholique de Louvain, Institute of Mechanics, Materials and Civil Engineering (IMMC), Avenue Georges Lemaître 4, B-1348 Louvain-la-Neuve, Belgium
[2]ETH Zürich, Institute for Atmospheric and Climate Sciences, Universitätstrasse 16, 8092 Zürich, Switzerland
[3]Vrije Universiteit Brussel, Department of Hydrology and Hydraulic Engineering, Pleinlaan 2, B-1050 Brussels, Belgium
[4]Université catholique de Louvain, Institute of Mechanics, Materials and Civil Engineering (IMMC) & Earth and Life Institute (ELI), Avenue Georges Lemaître 4, B-1348 Louvain-la-Neuve, Belgium
[5]Delft University of Technology, Delft Institute of Applied Mathematics (DIAM), Mekelweg 4, 2628CD Delft, The Netherlands

*Correspondence to:* Philippe Delandmeter (philippe.delandmeter@uclouvain.be)

**Abstract.** The discontinuous Galerkin (DG) finite element method is well suited to the modelling, with a relatively small number of elements, of three-dimensional flows exhibiting strong velocity or density gradients. Its performance can be highly enhanced by having recourse to r-adaptivity. Here, a vertical adaptive mesh method is developed for DG finite elements. This method, originally designed for finite difference schemes, is based on the vertical diffusion of the mesh nodes, with the diffusivity controlled by the density jumps at the mesh element interfaces.

The mesh vertical movement is determined by means of a conservative Arbitrary Lagrangian-Eulerian (ALE) formulation. Though conservativity is naturally achieved, tracer consistency is obtained by a suitable construction of the mesh vertical velocity field, which is defined in such a way that it is fully compatible with the tracer and continuity equations at a discrete level.

The vertically adaptive mesh approach is implemented in the geophysical and environmental flow model SLIM 3D (www.climate.be/slim). Idealised benchmarks, aimed at simulating the oscillations of a sharp thermocline, are dealt with. Then, the relevance of the vertical adaptivity technique is assessed by simulating thermocline oscillations of Lake Tanganyika. The results are compared to measured vertical profiles of temperature, showing similar stratification and outcropping events.

## 1 Introduction

The vertical discretisation strategy of marine models has evolved drastically during the last five decades. The first models were using $z$-coordinates (e.g. Bryan, 1969), discretising the ocean into fixed horizontal levels, resulting in a stepwise representation of the ocean bottom. Later, other discretisations were developed, mainly inspired by the progress in atmospheric modelling,





for which coordinates based on the pressure field were used (Sutcliffe, 1947; Eliassen, 1949; Phillips, 1957). In oceanography, such developments led to $\rho$- or $\sigma$- coordinates (Freeman et al., 1972; Owen, 1980; Bleck and Boudra, 1986; Nihoul et al., 1986; Blumberg and Mellor, 1987). For the $\rho$-coordinates, the vertical coordinate is based on the density field. This method is well suited to tracer transport in the ocean interior, which occurs mainly along isopycnal surfaces (Griffies et al., 2000). The $\sigma$-coordinates place a constant number of levels evenly spaced in the vertical column. This method enables a smooth
representation of the bottom topography, which is particularly appropriate for coastal applications. The discretisation of the internal pressure gradient constitutes a major difficulty of the vertical coordinate systems for which none of the iso-coordinate surface is horizontal, such as the $\sigma$-coordinate system. Since the iso-$\sigma$ surfaces are generally not horizontal, it is difficult to maintain a vertically-stratified water body at rest in a domain with a steep bottom slope. This problem has been extensively studied and documented (e.g. Haney, 1991; Deleersnijder and Beckers, 1992; Stelling and Van Kester, 1994; Mellor et al.,
10  1994, 1998).

Later on, the general $s$-coordinate system was developed (Gerdes, 1993a, b; Song and Haidvogel, 1994), which was also inspired by generalised coordinates in atmospheric modelling (Kasahara, 1974). The $s$-coordinates are able to arbitrarily combine $z$-, $\rho$- and $\sigma$- coordinates. With this technique, a single grid is able to cope with $z$-levels close to the ocean surface, $\rho$-levels in the interior and $\sigma$-levels close to the bottom. Those different methods have their own strengths and weaknesses, which are
discussed in details in Griffies et al. (2000) for large scale applications. The modeller is free to build an a-priori optimal mesh, depending on the application.

The $s$-coordinates reach their limits when the optimal vertical distribution of the mesh nodes should vary in space and time. This is why Burchard and Beckers (2004) proposed a non-uniform grid system that adapts the resolution by moving the nodes during the simulation so as to minimise a suitably-defined error measure (Hanert et al., 2006, 2007). This is referred to as
r-adaptive method, i.e. the mesh nodes are moved without modifying the mesh topology. Hofmeister et al. (2010) implemented this type of vertical movement of the mesh in a three-dimensional model. This method reduces the numerical mixing and the errors in the pressure gradient computation, enabling realistic simulations of large inflows in the Baltic sea (Hofmeister et al., 2011). The price of the reduced numerical mixing is the computation of the mesh velocity. Dealing with the mesh movement within the hydrodynamics equations has negligible extra cost. Indeed, free-surface models do already move the mesh to take
into account the surface motion (e.g. Shchepetkin and McWilliams, 2005).

This r-adaptive method only moves the nodes in the vertical direction. In contrast, the models using 3D hr-adaptation (e.g. Piggott et al., 2005, 2008; Hill et al., 2012) follow a completely different approach, based on tetrahedral meshes unstructured in the three directions.

To the best of authors' knowledge, the vertically adaptive coordinate method has only been applied to structured grid models.
The objective of this work is to adapt the method to an unstructured-mesh discontinuous Galerkin (DG) finite element model, namely the three-dimensional version of the Second-generation Louvain-la-Neuve Ice-ocean Model (SLIM 3D, www.climate. be/slim).

SLIM 3D is a baroclinic model for coastal flows that solves the 3D hydrostatic equations under the Boussinesq approximation (Blaise et al., 2010; Comblen et al., 2010; Kärnä et al., 2013). The model is based on the DG finite element method. The





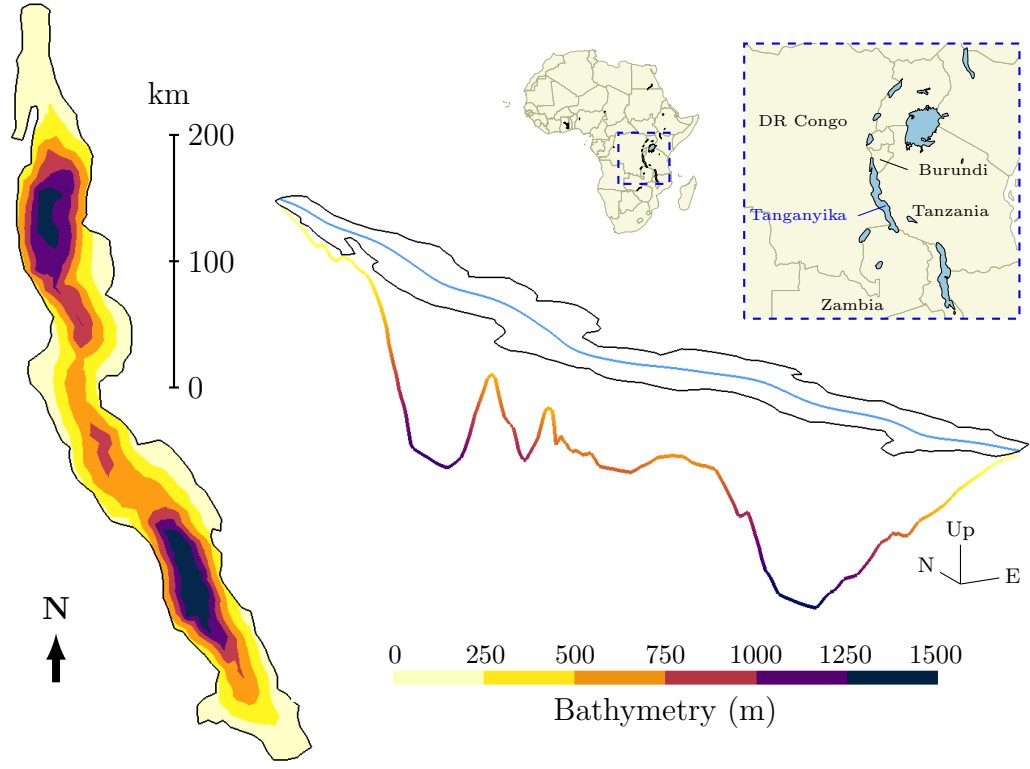

**Figure 1.** Lake Tanganyika bathymetry (Naithani et al., 2002): isobaths (left) and perspective view of bottom profile along the main axis of the lake (right).

latter is well suited for advection-dominated problems (Bassi and Rebay, 1997; Cockburn et al., 2000; Bernard et al., 2007) exhibiting strong gradients of the solution. Furthermore, it has different advantages, such as local and global conservativity, or the compactness of the stencil, which enables an easy and efficient parallel implementation (Seny et al., 2013, 2014). The inter-element discontinuities of the solution constitute a good estimate of the discretisation error (Ainsworth, 2004; Bernard et al., 2007). Previous applications of SLIM 3D have focused on coastal flows, estuaries and river plume dynamics, where a high resolution is required in the surface layer which is under the direct influence of the wind stress. Accordingly, the mesh resolution is increased close to the surface (Delandmeter et al., 2015).

In this work, the model is applied to Lake Tanganyika, especially its thermocline movement, for which the depth and location where high resolution is desirable vary in time. Lake Tanganyika is the largest of the East African Great Lakes in terms of water volume and the second largest in terms of surface (Ogutu-Ohwayo et al., 1997). It is shared by four countries: Burundi, Democratic Republic of the Congo, Tanzania and Zambia (Fig. 1). Schematically, the waters of the lake exhibit two layers separated by a thermocline (Coulter and Spigel, 1991; Naithani and Deleersnijder, 2004). The dynamics of the lake thermocline differs between the dry wind season and the wet season during which the wind stress is significantly smaller.





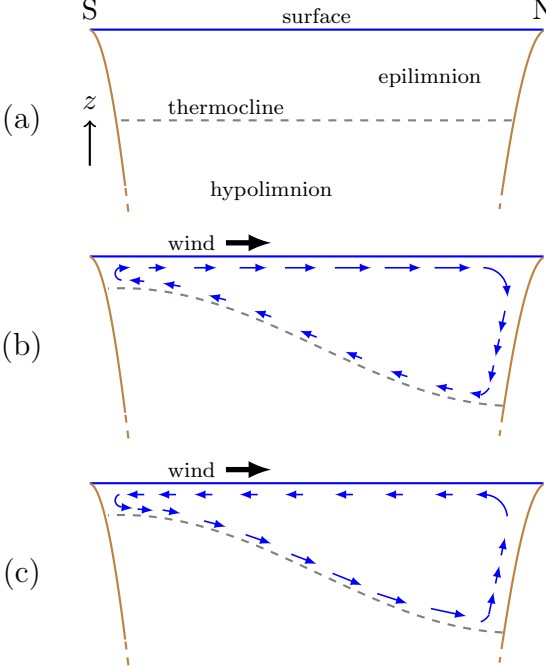

**Figure 2.** Schematical lake configuration with no wind stress (a) and with the south-easterly wind during the dry season in line with the classical circulation (b, Mortimer, 1961) such that the surface water follows the wind, or according to the non-classical circulation (c, Verburg et al., 2008) with reversed currents.

A few 3D modelling studies were conducted on the lake. Huttula (1997) used a 3D barotropic model to simulate the transport of sediment in sub-regions of the lake. Podsetchine et al. (1999) used the same model to study the lake response to different wind stress regimes. Their simulations were focused on the diurnal cycle of the water velocity. They did not aim to model the seasonal variability of the current and the thermocline dynamics, for which the 3D model must be baroclinic.

Verburg et al. (2008) studied the overturning circulation in the lake. In contrast to the classical overturning circulation in a two-layer lake under constant wind stress (Fig. 2, Mortimer, 1961), they proposed a reversed circulation with the deepest water of the epilimnion following the wind and the surface water flowing in the opposite direction. Verburg et al. (2011) quantified the conditions for which they proposed a counter-wind surface circulation, driven by the surface heat flux of the lake. They used a 3D model (Hodges et al., 2000) to assess their hypothesis by simulating the lake dynamics in 1996.

This paper presents the development of a vertically adaptive coordinate system for SLIM 3D and its validation on simple benchmarks. The improved model is then applied to Lake Tanganyika to investigate the adaptive coordinates efficiency in a realistic application. In Section 2, the vertical adaptive mesh approach is defined, and a new computation of the mesh velocity is designed, which guarantees tracer conservation and consistency. The formulation of the internal pressure gradient is detailed. Finally, information about the input data used for the Lake Tanganyika simulation is provided. In Section 3, the method

performance is evaluated on two benchmarks: the internal seiche and the steady state thermocline position under a constant





wind stress. A convergence analysis is performed. Then the model is applied to the oscillations of the thermocline in Lake Tanganyika. The results are discussed in Section 4 and conclusions are drawn in Section 5.

## 2  Methods

### 2.1  Numerical modelling

In Kärnä et al. (2013), SLIM 3D equations are discretised on a mesh that uniformly moves in the vertical direction to follow the free surface movement (Campin et al., 2004). In the present study, an additional vertical mesh velocity is developed to obtain the desired mesh vertical resolution. In order to satisfy the crucial properties of both mass conservation and consistency, the moving mesh velocity is computed differently from Kärnä et al. (2013) and is based on the discrete formulation of the continuity and tracer equations. Indeed, the moving mesh velocity used in previous versions of SLIM 3D was suffering from small errors for the tracer consistency preservation: a constant concentration did not remain constant while transported. Those errors were getting worse with the new vertical adaptive method. The computation of the internal pressure horizontal gradient is also described.

#### Moving mesh and ALE formulation

The model deals with the horizontal momentum, temperature and tracer equations using the Arbitrary Lagrangian-Eulerian (ALE) formulation (Formaggia and Nobile, 2004). Those equations are solved in a moving domain $\Omega_t$. Considering the fixed domain $\tilde{\Omega}$, $\mathcal{A}$ is an invertible mapping from $\tilde{\Omega}$ to $\Omega_t$ (similar to Deleersnijder and Ruddick, 1992):

$$\mathcal{A} : \tilde{\Omega} \longrightarrow \Omega_t, \quad \mathcal{A}\big(\tilde{x}, \tilde{y}, \tilde{z}, \tilde{t}\big) = \big(x, y, z, t\big), \tag{1}$$

$$\begin{cases} x = \tilde{x} \\ y = \tilde{y} \\ z = z(\tilde{x}, \tilde{y}, \tilde{z}, \tilde{t}) \\ t = \tilde{t}. \end{cases}$$

The vertical moving mesh velocity $w_m$, which denotes the movement of the moving domain $\Omega_t$ with respect to the fixed domain $\tilde{\Omega}$, reads in the ALE framework:

$$w_m = \frac{\partial z}{\partial \tilde{t}}. \tag{2}$$

#### A conservative and consistent moving mesh

Let $\boldsymbol{u}$ and $w$ denote the horizontal and the vertical water velocities, respectively. The temperature $T$ equation reads in the moving domain $\Omega_t$ using the Eulerian coordinates:

$$\frac{\partial T}{\partial t} + \boldsymbol{\nabla}_h \cdot (\boldsymbol{u}T) + \frac{\partial(wT)}{\partial z} = D_T, \tag{3}$$





where $\boldsymbol{\nabla}_h \cdot (\boldsymbol{u}T) = \frac{\partial(uT)}{\partial x} + \frac{\partial(vT)}{\partial y}$ and $D_T$ represents the diffusive terms of the temperature equations.

The Jacobian of the mapping (Eq. 1) is $J = \partial z / \partial \tilde{z}$. Following the steps defined in Formaggia and Nobile (2004), the temperature equation transforms to:

$$\frac{1}{J} \frac{\partial(JT)}{\partial \tilde{t}} + \boldsymbol{\nabla}_h \cdot (\boldsymbol{u}T) + \frac{\partial((w - w_m)T)}{\partial z} = D_T, \tag{4}$$

where the time derivative is calculated on the fixed mesh. This conservative formulation preserves the total heat by construction in a finite element scheme (Formaggia and Nobile, 2004).

Eq. 4 is then discretised using the finite element formalism. The diffusive terms $D_T$, detailed in Kärnä et al. (2013), are zero as long as temperature is constant and, hence, are ignored hereinafter. The DG approximation $\hat{T}$ of $T$ is the interpolation of the discontinuous nodal values $T_i$ using the usual P1 shape functions $\phi_i$, bi-linear in the case of a mesh composed of extruded
triangles and tri-linear in the case of extruded quads:

$$\hat{T} = \sum_i \phi_i T_i. \tag{5}$$

The fields $\boldsymbol{u}$, $w$ and $w_m$ are discretised similarly. For clarity, the hat sign is removed from the notation hereinafter. The weak DG formulation of Eq. 4 reads:

$$\frac{d}{dt}\left\langle JT\phi_j \right\rangle \quad - \left\langle JT\boldsymbol{u} \cdot \boldsymbol{\nabla}_h \phi_j \right\rangle + \left\langle\!\left\langle JT^*(\boldsymbol{u}^* \cdot \boldsymbol{n}_h)\phi_j \right\rangle\!\right\rangle_l + \left\langle\!\left\langle JT^*(\boldsymbol{u}^* \cdot \boldsymbol{n}_h)\phi_j \right\rangle\!\right\rangle_h$$

$$- \left\langle JT(w - w_m)\frac{\partial \phi_j}{\partial z} \right\rangle + \left\langle\!\left\langle JT^*(w^d - w_m^d)n_z \, \phi_j \right\rangle\!\right\rangle_h = 0, \qquad \forall j. \tag{6}$$

$\left\langle \cdot \right\rangle$, $\left\langle\!\left\langle \cdot \right\rangle\!\right\rangle_l$ and $\left\langle\!\left\langle \cdot \right\rangle\!\right\rangle_h$ are the integral over the domain, the lateral and the horizontal interfaces, respectively. All those integrals are computed over the fixed domain $\tilde{\Omega}$. At the interfaces, the velocity $\boldsymbol{u}^*$ is evaluated with an approximated Riemann Solver (Kärnä et al., 2013), $w^d$ and $w_m^d$ are the values of $w$ corresponding to the lower element and $T^*$ is the temperature taken from the upstream element. $\boldsymbol{n}_h$ and $n_z$ refer to the horizontal and vertical components of the normal vector to the interfaces.

In Kärnä et al. (2013), the discrete moving mesh velocity $w_m$ is obtained by interpolating Eq. 2 at nodes. However, this approach breaks the consistency. In this work, the mesh velocity is constructed from the position of the new mesh in a way that ensures tracer consistency at a discrete level. The consistency of the method relies on the compatibility of the tracer equation (Eq. 4) with the continuity equation:

$$\boldsymbol{\nabla}_h \cdot \boldsymbol{u} + \frac{\partial w}{\partial z} = 0, \tag{7}$$

whose weak formulation reads:

$$-\left\langle w \frac{\partial \phi_j}{\partial z} \right\rangle + \left\langle\!\left\langle w^d \phi_j \, n_z \right\rangle\!\right\rangle_h =$$
$$\left\langle \boldsymbol{u} \cdot \boldsymbol{\nabla}_h \phi_j \right\rangle - \left\langle\!\left\langle \boldsymbol{u}^* \cdot \boldsymbol{n}_h \, \phi_j \right\rangle\!\right\rangle_l - \left\langle\!\left\langle \boldsymbol{u}^* \cdot \boldsymbol{n}_h \, \phi_j \right\rangle\!\right\rangle_h, \quad \forall j. \tag{8}$$

Inserting Eq. 8 into Eq. 6 and assuming that $T$ is constant, one obtains the following equality which, if satisfied for $w_m$, guarantees the consistency:

$$\frac{d}{dt}\left\langle J\phi_j \right\rangle = -\left\langle Jw_m \frac{\partial \phi_j}{\partial z} \right\rangle + \left\langle\!\left\langle Jw_m^d n_z \, \phi_j \right\rangle\!\right\rangle_h, \qquad \forall j. \tag{9}$$



To solve Eq. 9, the equation is here integrated in time using an explicit Euler time scheme (other time schemes follow a similar development):

$$\frac{\left\langle \phi_j \right\rangle^{n+1} - \left\langle \phi_j \right\rangle^n}{\Delta t} = -\left\langle w_m \frac{\partial \phi_j}{\partial z} \right\rangle^n + \left\langle\!\left\langle w_m^d n_z\, \phi_j \right\rangle\!\right\rangle_h^n, \qquad \forall j. \tag{10}$$

It is noteworthy that the Jacobian $J$ which appeared in Eq. 9 is not present in Eq. 10. This is because while the integrals were computed on the fixed domain $\tilde{\Omega}$ in Eq. 9, they are computed on the moving domain in Eq. 10. The mesh on which the integral is computed is referred to by using the superscripts $n$ or $n+1$.

Starting from the bottom boundary condition $w_m|_{z=-h} = 0$, Eq. 10 can be integrated element by element from bottom to top to obtain the moving mesh velocity. Note that, in the interface integrals, the normal $n_z$ is pointing outward.

Using this method, the temperature equation reduces by construction to the continuity equation if the temperature is constant, and therefore the consistency property holds valid. Salinity and tracer equations follow exactly the same scheme.

**Internal pressure gradient**

In finite difference models using terrain following meshes, the computation of the horizontal gradient of the internal pressure gradient is complex. Considerable efforts were made to reduce the errors in this computation and to limit the spurious pressure gradient (e.g. Thiem and Berntsen, 2006; Berntsen and Oey, 2010; Berntsen et al., 2015). For the case of vertically adaptive meshes, Gräwe et al. (2015) showed that the internal pressure gradient problem is reduced because of the horizontal smoothing of the mesh.

The pressure gradient formulation in SLIM 3D is different from the finite difference schemes, the equations are in $z$-coordinates and not in $\sigma$- or $s$- coordinates. The fact that the levels are not horizontal is handled by the finite element formulation. However, having steep slope also induces difficulties. The weak formulation of the horizontal gradient of a field $f$ reads:

$$\left\langle \boldsymbol{\nabla}_h f \cdot \boldsymbol{\phi}_j \right\rangle = -\left\langle f\, \boldsymbol{\nabla}_h \cdot \boldsymbol{\phi}_j \right\rangle + \left\langle\!\left\langle f \boldsymbol{\phi}_j \cdot \boldsymbol{n}_h \right\rangle\!\right\rangle_h + \left\langle\!\left\langle f \boldsymbol{\phi}_j \cdot \boldsymbol{n}_h \right\rangle\!\right\rangle_l, \quad \forall j \tag{11}$$

Since first order shape functions are used in the finite element formulation, only first order polynomials are interpolated exactly at the integration points in order to compute the integrals of the right hand side of Eq. 11.

In the previous version of SLIM 3D (Kärnä et al., 2013), the internal pressure gradient was obtained by first integrating the density deviation $\rho' = \rho - \rho_0$, with $\rho$ and $\rho_0$ the water density and the water reference density, and then computing its horizontal gradient:

$$\frac{1}{\rho_0} \boldsymbol{\nabla}_h p = \frac{g}{\rho_0} \boldsymbol{\nabla}_h \underbrace{\int_z^\eta \rho' d\zeta}_{r}. \tag{12}$$

Even for a vertically linear $\rho'$, $r$ is a quadratic function that cannot be represented exactly. Those integration errors generate errors in the gradient $\boldsymbol{\nabla}_h p$.



The new approach consists in computing the horizontal derivative before the vertical integration:

$$\frac{1}{\rho_0}\boldsymbol{\nabla}_h p = \frac{g}{\rho_0}\int_z^\eta \boldsymbol{\nabla}_h \rho' d\zeta + \frac{g}{\rho_0}\rho'|_\eta \boldsymbol{\nabla}_h \eta. \tag{13}$$

Using Eq. 13, the computation of the gradient of a linearly stratified sea does not involve a quadratic field, and the computed internal pressure gradient is zero, up to the machine accuracy.

## 5 Vertical adaptive mesh velocity

To obtain accurate results at a reasonable computational cost, the mesh vertical resolution should be high in areas with strong stratification or shear, and low elsewhere. In this study, the refinement is achieved as a function of the stratification only. The shear is thus ignored, but it could be taken into account similarly to what is developed hereinafter for the stratification. The mesh velocity reads:

$$w_m = \frac{h+z}{h+\eta}\frac{\partial \eta}{\partial t} + w^*. \tag{14}$$

The first term of the right hand side is the mesh velocity due to the free surface movement while the second term is the mesh adaptive velocity.

The mesh resolution variation results from a diffusion process of the Eulerian vertical coordinate $z(\tilde{x},\tilde{y},\tilde{z},\tilde{t})$. In contrast to Burchard and Beckers (2004) and Hofmeister et al. (2010), the diffusion process is not defined in the continuous domain but directly on the discrete mesh. The mesh is made up of vertically extruded triangles or quads to form columns of prisms. As a consequence, the mesh also consists of vertical columns of nodes, which are the vertices of the prisms. Those nodes are connected by vertical segments. For each column composed of $n$ vertical segments, the nodes are labelled with an index $i$ varying between 0 at the bottom and $n$ at the top. Segment $i+1/2$ joins nodes $i$ and $i+1$. The objective of the mesh adaptation is to distribute the nodes of a column such that:

$$h_{i+1/2}\, e_{i+1/2} = \text{constant}, \tag{15}$$

where $h_{i+1/2} = (z_{i+1} - z_i)$ is the segment height and $e_{i+1/2}$ is a relevant measure of the segment error. A finite difference diffusion equation is implemented:

$$\begin{cases} w_i^* &= \kappa_{z\,i+1/2}(z_{i+1} - z_i) - \kappa_{z\,i-1/2}(z_i - z_{i-1}), \quad \forall i = 1\ldots n-1, \\ w_0^* &= w_n^* = 0. \end{cases} \tag{16}$$

The mesh "diffusivity" $\kappa_z$ (which has the physical unit $\mathrm{s}^{-1}$) is then defined by:

$$\kappa_{z\,i+1/2} = \frac{e_{i+1/2}^2 + f_{e\,i+1/2}}{\tau}. \tag{17}$$

The dependency in $e^2$ ensures that, at a given level, if the upper segment error is larger than the lower segment error, the diffusion process will tend to reduce the upper segment size, and vice versa. The background diffusivity $f_e\,\tau^{-1}$ allows for





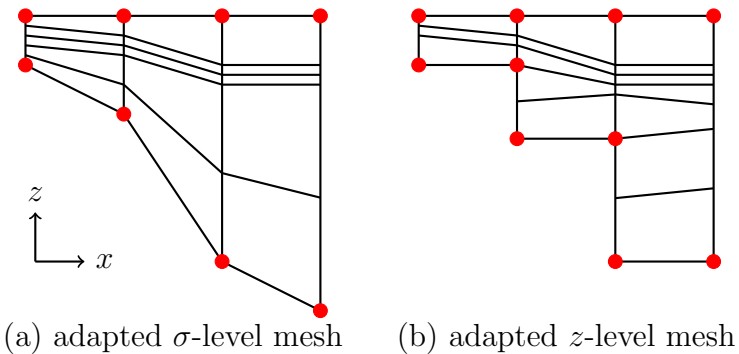

(a) adapted $\sigma$-level mesh  (b) adapted $z$-level mesh

**Figure 3.** Comparison between an adapted $\sigma$-level mesh (a) and an adapted $z$-level mesh (b). The nodes marked with a red dot are fixed during the mesh vertical diffusion process, which is independent of the movement necessary to accommodate to the motion of the free surface (Eq. 14).

manual control of the mesh resolution independently of the discretisation error. The time $\tau$ is a relaxation parameter controlling the speed of the adaptivity process.

The main difference between the original approach and the implementation in SLIM 3D is the definition of the error density $e^2$. In Hofmeister et al. (2010), a function of shear and stratification is used. In DG finite element methods, the discretisation

error converges at the same rate as the inter-element discontinuities (jumps) of the solution (Ainsworth, 2004; Bernard et al., 2007). The error density is then defined as a function of the vertical jumps:

$$e^2_{i+1/2} = \frac{[\rho]^2_i + [\rho]^2_{i+1}}{(\Delta\rho)^2}, \qquad (18)$$

where $[\rho]_i$ refers to the maximum jump, between all the upper DG values adjacent to continuous node $i$ and lower DG values adjacent to this same node.

This diffusion algorithm is valid for meshes with both $\sigma$- and $z$- levels. On a mesh with $\sigma$-levels, the number of levels is constant over the entire mesh and the bathymetry is continuous (Fig. 3a). On a mesh with $z$-levels, the number of levels is not constant and the bathymetry is discontinuous (Fig. 3b).

The discrete mesh is updated by interpolating the function defined in Eq. 14 at the element vertices. Then the $z$-coordinates of the vertices are smoothed in the horizontal direction, except on the mesh lateral boundaries. This smoothing is achieved with

a simple 2-step algorithm (Fig. 4). First, for each element of the mesh, the $z$-coordinates of the upper and lower nodes are set to the mean value of those upper and lower nodes, respectively. The $z$ field is now discontinuous. Second, $z$ is projected onto a continuous field, such that the mean value of every vertex is preserved. This simple algorithm smooths the $z$-coordinates in the horizontal direction. Eventually, $z$ is corrected such that all the elements have a thickness between appropriate minimal and maximal values. This correction is achieved by a double loop over each segment column. First, looping from bottom to top,

the top node of each segment is moved if necessary. Second, looping from top to bottom, the bottom node of each segment is moved.





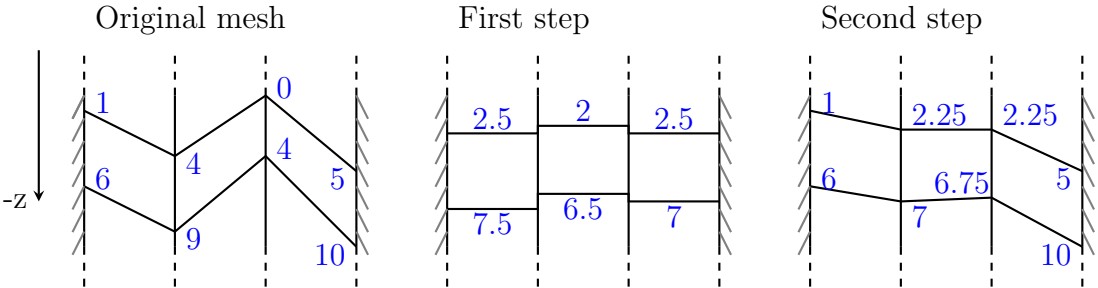

**Figure 4.** Illustration of the horizontal smoothing algorithm on a 2D "$x$-$z$" mesh. First, for each element, the upper and lower nodal values of the vertical coordinate $z$ are set to their mean value, then $z$ is projected onto a continuous field.

## 2.2 Thermocline oscillations of Lake Tanganyika model set-up

**Summary of Lake Tanganyika dynamics**

Lake Tanganyika is very long ($\sim$ 650 km in length), narrow ($\sim$ 50 km wide on average) and deep, with an average depth of 570 m and a maximum depth of 1470 m (Fig. 1), making it the second deepest lake in the world. The two layers composing the
lake are called the epilimnion and the hypolimnion. The epilimnion, which is the shallow upper layer, is relatively warm (24-28°C,  Coulter and Spigel, 1991; Naithani et al., 2003) and has a typical depth of about 50 m. Below the thermocline, the deep hypolimnion, is composed of cooler water ($\sim$ 23.5°C). Forced by the surface wind stress, the thermocline oscillates. There are two main seasons in the region. During the dry season, approximately from April or May to September, strong south-easterly wind blows along the main axis of the lake (Docquier et al., 2016). The wind pushes the epilimnion water towards the north,
causing upwelling at the southern tip and downwelling at the northern tip (Fig. 2), and resulting in a thermocline tilted towards the north. The wind oscillations are characterised by a period of 3 to 4 weeks (Naithani et al., 2002), which is of the same order of magnitude as the period of the first free mode of oscillation of the thermocline (Naithani et al., 2003), giving rise to quasi-resonance. Thus, during the abovementioned season, the thermocline oscillations are essentially a direct response to the wind forcing, i.e. large-amplitude, forced oscillations (Gourgue et al., 2011). The thermocline is deep and sharp at the northern
tip but becomes diffuse at the southern tip (Coulter and Spigel, 1991). During strong wind events, the oscillations are so large that the thermocline outcrops. Besides the hydrodynamical effect, the mixed layer also deepens during the dry season. This mixing is due to evaporative-driven cooling (Thiery et al., 2014a, b). On the other hand, during the wet season (from October to March), the wind stress is significantly smaller, leading to thermocline oscillations that may be viewed as progressively decaying internal seiches.

**Wind forcing**

Two wind datasets are available: a time series of measurements at one location, and a spatial wind map obtained from a model.




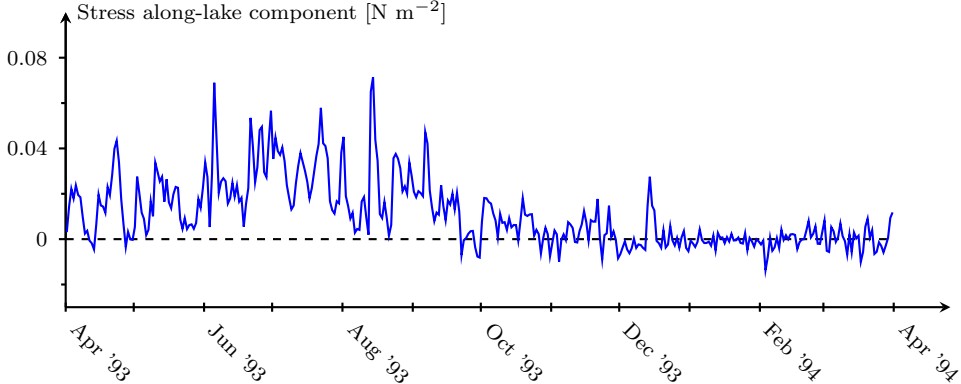

**Figure 5.** Along-lake component of the daily-averaged wind stress measured in Mpulungu, at the southern tip of the lake. Positive values mean northwestward blowing wind (Huttula et al., 1999).

Wind speed and direction were measured every hour from April 1993 to August 1994 (Huttula et al., 1999) in Mpulungu, at the southern tip of the Lake ($8°45'$S; $31°6'$E). These wind speed observations were used for earlier studies using a 2D reduced gravity model (Naithani et al., 2003, 2007; Naithani and Deleersnijder, 2004; Gourgue et al., 2007, 2011). Figure 5 shows the daily-averaged wind stress along the main axis of the lake.

On the other hand, non-uniform wind data were obtained from the COSMO-CLM$^2$ model, which couples the non-hydrostatic regional climate model COSMO-CLM version 4.8 to the Community Land Model version 3.5 (CLM3.5) and the Freshwater Lake model (FLake; Davin and Seneviratne, 2012). The COSMO-CLM$^2$ model was recently applied in its tropical configuration (Akkermans et al., 2014; Panitz et al., 2014) to assess the two-way interactions between the African Great Lakes and the surrounding climate (Thiery et al., 2015, 2016; Docquier et al., 2016) as well as evaluate natural hazards in the region
(Jacobs et al., 2016a, b). The climate simulations were conducted at a horizontal resolution of $0.0625°$ ($\sim 7$ km) for the period 1996-2008 and provide near-surface wind fields at a temporal resolution of 3 h.

Figure 6 shows the component of the surface wind stress along the main axis of the lake at three locations. While the wind is mostly blowing north-westward during the dry season (April - September), it is much weaker and does not have a dominant direction during the wet season. The wind is weaker in the northern part of the lake (Docquier et al., 2016).

**Model set-up**

Two configurations of the model are run. In the first configuration, aimed to analyse the effect of adaptive coordinates, no vertical diffusivity is applied to the temperature field, such that the modelled thermocline should remain sharp. The vertical viscosity is determined from the $\kappa$-$\epsilon$ turbulence closure model implemented in GOTM (Burchard et al., 1999) and coupled to SLIM 3D (Kärnä et al., 2012). Moreover, the homogeneous wind stress from Mpulungu measurement is applied at the entire
lake surface and no surface heat flux is applied.





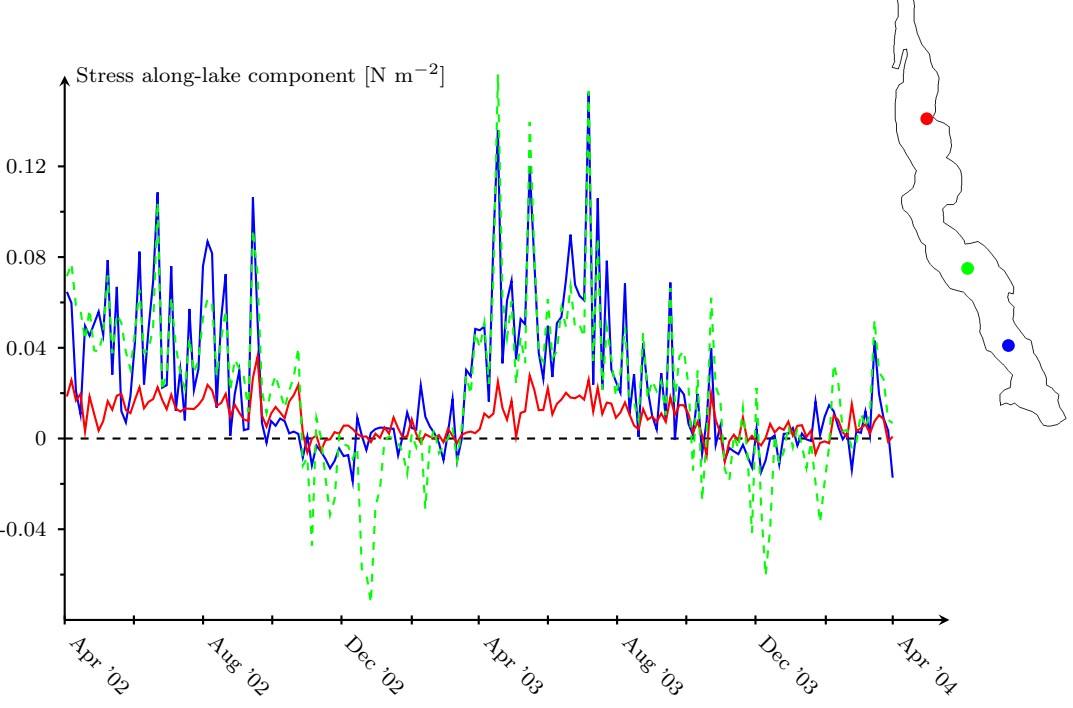

**Figure 6.** Along-lake component of the daily-averaged wind stress, at the northern, central and southern part of the lake, between April 2002 and 2004, as modelled by the regional climate model COSMO-CLM$^2$ (Thiery et al., 2015).

In the second set-up, the vertical diffusivity and viscosity are taken into account. They are determined from the $\kappa$-$\epsilon$ turbulence closure model. The spatial wind from COSMO-CLM$^2$ is applied. The surface heat flux is parameterised by adding a relaxation term, in the upper layer of the lake. The relaxation term is defined as:

$$f_{\text{relax}} = \frac{\max(z_r + z, 0)}{z_r \tau_r}(T_{ref} - T),$$

(19)

5   with $T_{ref}$ the reference temperature, $z_r$ the depth of the relaxation zone and $\tau_r$ the relaxation time parameter. The surface reference temperature comes from the same data set as the wind data, i.e. from the COSMO-CLM$^2$ model. Furthermore, the depth of the relaxation zone is set to $z_r = 12$ m, which corresponds to a typical value for the photic depth (Descy et al., 2006), which is used as a proxy for the water column influenced by solar radiation and other heat fluxes. The relaxation time is equal to $\tau_r = 10$ days, which was obtained after calibration. Figure 7 illustrates the evolution of this temperature at two locations:

10   Kigoma, in the northern basin, and Mpulungu, at the southern tip of the lake. It is observed that at both locations the surface temperature drops during the dry season.





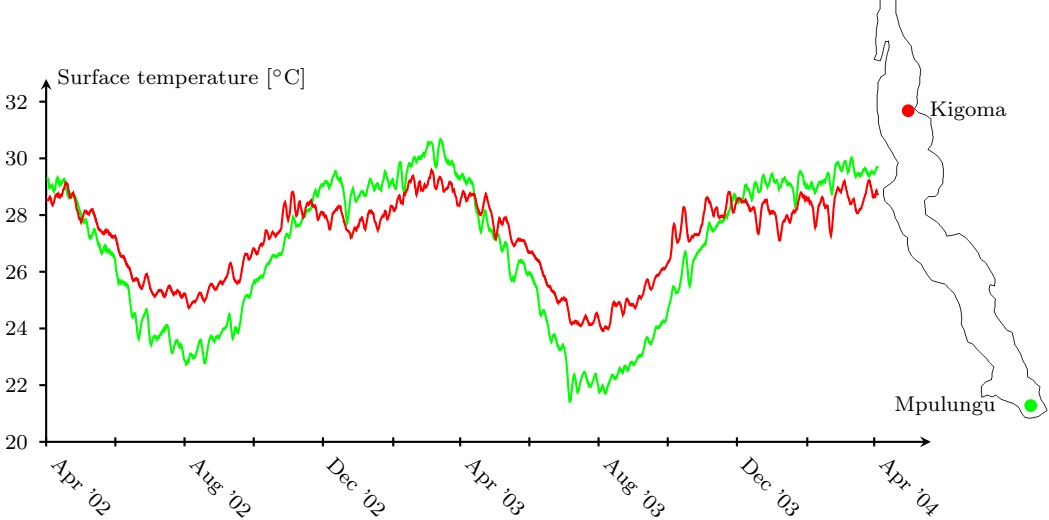

**Figure 7.** Time series of the surface temperature at Kigoma (red curve) and Mpulungu (green curve), as modelled by COSMO-CLM[2] (Thiery et al., 2015).

**Lake temperature vertical profile**

Within the framework of the CLIMLAKE project (Descy et al., 2006), the water temperature was measured at Kigoma and Mpulungu (see location on Fig. 7) between 2002 and 2004, at vertical intervals of 20 metres between 0 and 100 m deep. This vertical profile temporal series is used for model validation. It is noteworthy that the surface temperature at Kigoma and

5  Mpulungu from the CLIMLAKE in situ measurements exhibit significant discrepancies with the data from the COSMO-CLM[2] data set, which are used as input data for the model simulations.

## 3 Results

Before applying SLIM 3D to Lake Tanganyika, the model is evaluated on simpler test cases. First the internal seiche benchmark of Hofmeister et al. (2010) is used to assess the model ability to preserve a sharp interface. Second, a convergence analysis is

10  performed on this benchmark. Third, the accuracy of the steady-state thermocline position under constant wind stress forcing is evaluated. Then, preliminary simulations of the Lake Tanganyika hydrodynamics are undertaken. For those runs, no vertical diffusivity is applied to the temperature field, and the wind stress measured at Mpulungu is forced uniformly at the lake surface. Adaptive and fixed meshes are compared on this set-up for both a 2D "$x$-$z$" and a 3D models. Eventually, the complete model for Lake Tanganyika is run.



## 3.1 Internal seiche

The first test to evaluate the adaptive coordinate system is the internal seiche modelling of Hofmeister et al. (2010). This two-layer benchmark bears similarities with the dynamics of the thermocline of Lake Tanganyika. It is fully defined in Hofmeister et al. (2010). The objective is to simulate the oscillations of the interface in a long (64-km long) and shallow (20-m deep) channel. In contrast to Lake Tanganyika, the density is here a function of salinity, not of temperature. Also, since the original test case is defined using a two-layer model, there is no vertical mixing. The goal is therefore to diffuse the interface as little as possible.

For this application, the error measure used to diffuse vertically the mesh is a function of the vertical jumps in the density field, with a small background error $f_e = 10^{-5}$. The time constant is set to $\tau = 100$ s, and the minimum and maximum heights of an element are set to 0.1 and 1.5 m, respectively.

Figure 8 compares the salinity vertical profile after half an oscillation for a fixed mesh (a) and the adaptive mesh (b), both with 20 levels. For both runs, the initial mesh is set such that it captures perfectly the interface initial position. While the fixed mesh induces large numerical mixing in one case, the mesh adapts and follows perfectly the interface owing to the vertical adaptivity in the other case.

## 3.2 Convergence analysis

To evaluate the model accuracy, a convergence analysis is performed for the internal seiche. The evolution of the interface depth at the right boundary of the domain is compared for different simulations using a number of fixed levels, varying between 10 and 320, which induces a level thickness varying between 2 m and 6.25 cm, using the same time step for all the simulations ($\Delta t = 60$ s). Two simulations are also performed using adaptive meshes, with 6 and 20 levels, respectively. In the coarse-resolution simulations, the interface is diffused (Fig. 8), and the seiche oscillates too slowly compared to the 320-level run. While the first oscillation is rather well captured by all the simulations, the coarse-resolution simulations miss the correct dynamics during the next oscillations. After two oscillations, the simulation with 20 fixed levels fails to reproduce the dynamics (Fig. 9). In contrast, the simulation with 20 adaptive levels is as accurate as the simulation with 320 fixed levels. The minimal number of adaptive levels for an acceptable simulation is 6. With this number, two thin levels stick to the upper part of the interface and two other on the lower part, while the two last levels cover the remaining part of the domain (one on the top and one on the bottom). The 6-level simulation with the adaptive method thereby produces results as good as the simulation with 80 fixed levels. Considering that the CPU time of the simulations is proportional to the number of levels (Fig. 10) and that the computational overhead of adaptive levels is negligible, the adaptive method is about 16 times faster for a similar accuracy.

## 3.3 Steady-state thermocline position under constant wind stress

To assess the model, the equilibrium position of the thermocline under a constant wind stress is evaluated in a 2D "$x$-$z$" domain. The position of the thermocline is approximated using the analytical solution of a 1D two-layer model, which simulates the epilimnion and hypolimnion vertically averaged velocity and the thermocline depth (Cushman-Roisin and Beckers, 2011). At



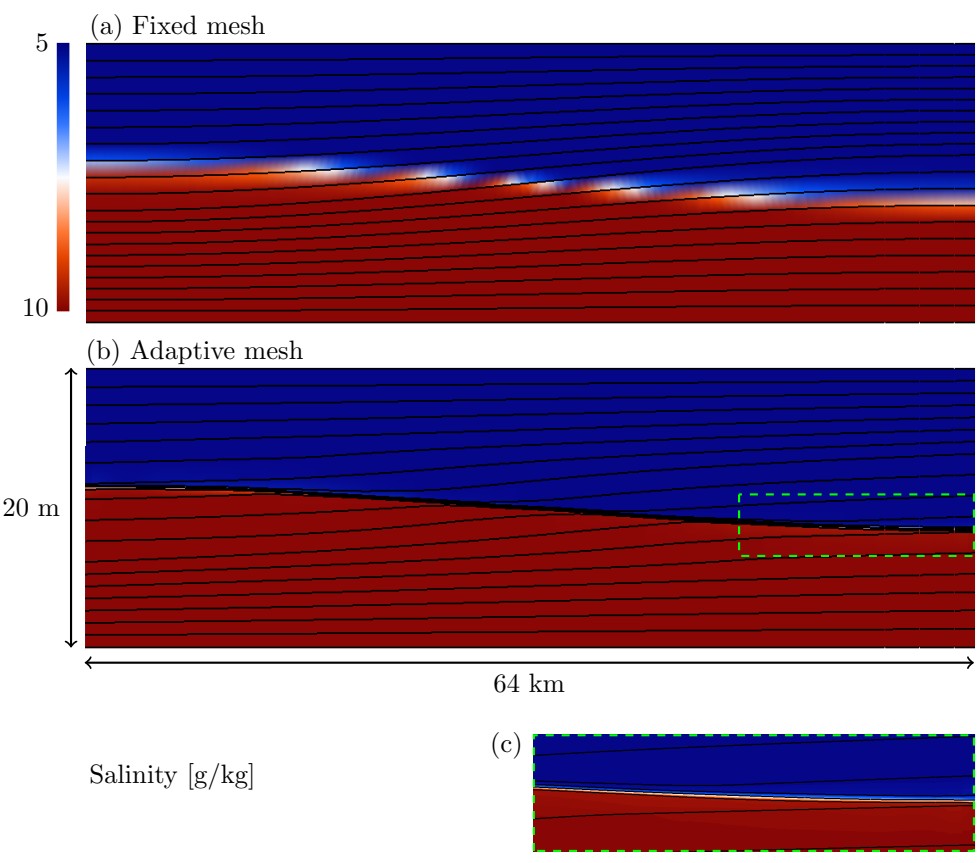

**Figure 8.** Comparison between the internal seiche modelling with (a) fixed and (b) adaptive meshes. While the first one induces important numerical mixing at the interface, the interface remains sharp in the second one. (c) Zoom of the area delimited by the green rectangle in (b), showing two levels with the highest vertical permitted resolution on both sides of the interface.

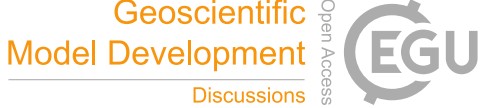



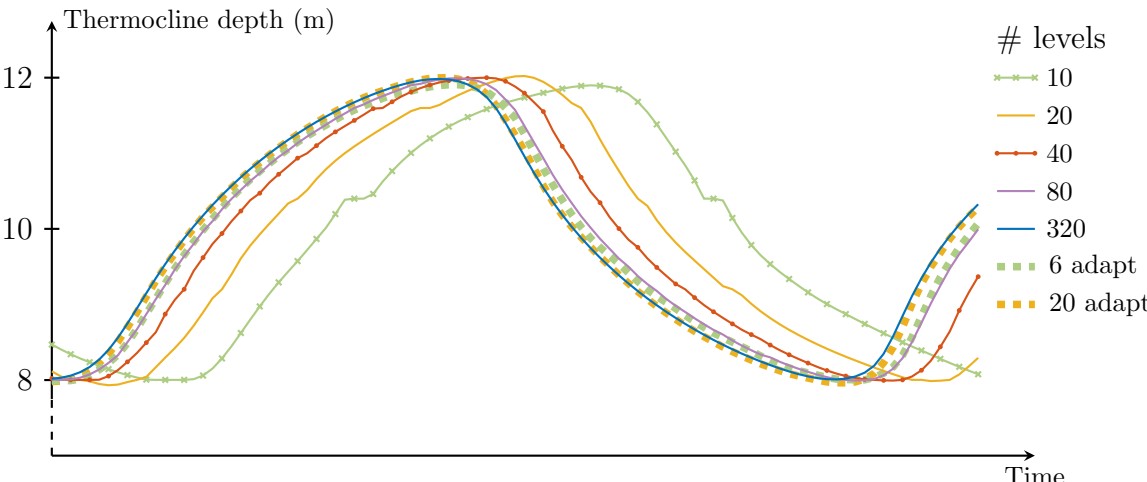

**Figure 9.** Convergence analysis for the internal seiche modelling. The graph shows the third oscillation of the seiche, at which moment the results are beginning to diverge from the high-resolution solution.

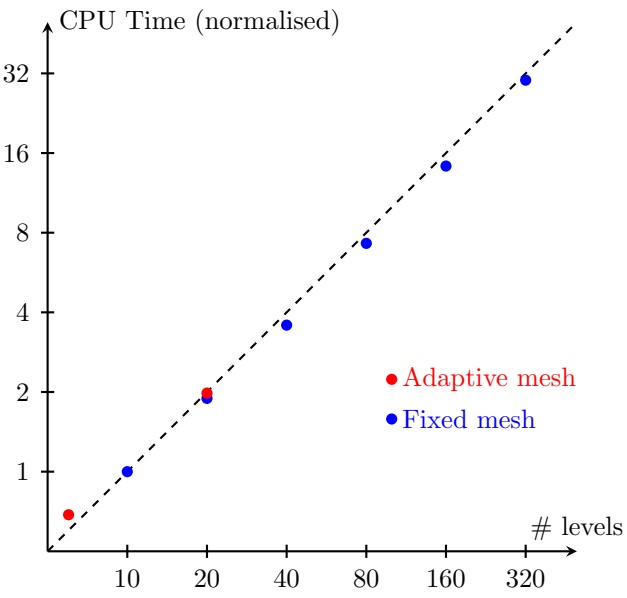

**Figure 10.** CPU time of the simulations for a different number of levels (using a loglog scale). CPU time are normalised by the 10 fixed levels simulation time. As expected, CPU time is proportional to the number of levels. Moreover, the computational overhead of adaptive levels is negligible.





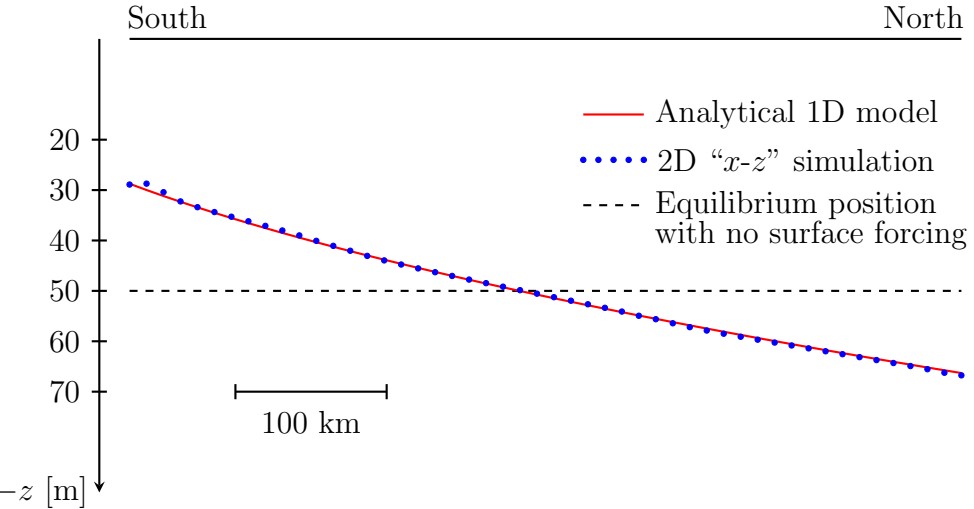

**Figure 11.** Comparison between the analytical steady state thermocline profile using the 1D two-layer approximation and the 2D "$x$-$z$" simulation, under a constant wind stress. The dashed line represents the thermocline position without wind stress.

the steady-state equilibrium, the pressure gradient due to the slope of the thermocline depth $d$ is balanced by the wind stress:

$$\epsilon g d \frac{\partial d}{\partial x} = \frac{\tau_x}{\rho}, \tag{20}$$

where $\epsilon = (\rho - \rho_0)/\rho_0$ is the relative density difference between the upper and the lower layers, $\rho$ and $\rho_0$ being the epilimnion and hypolimnion densities, respectively. $g$ is the gravitational acceleration and $\tau_x$ the wind stress. The solution of Eq. 20 reads:

$$d = \sqrt{A + \frac{2\tau_x}{\rho \epsilon g} x}, \tag{21}$$

where constant $A$ is such that the total epilimnion volume is conserved:

$$\int_0^L \sqrt{A + \frac{2\tau_x}{\rho \epsilon g} x} \, \mathrm{d}x = d_0 L. \tag{22}$$

$d_0$ is the initial epilimnion height (when the thermocline is horizontal) and $L$ is the length of the lake.

10  The thermocline depth is simulated for a wind stress of 0.02 N/m², for which there is no outcropping. The simulation starts with the thermocline located at $d_0 = 50$ m and the model runs until a steady-state is arrived at, resulting in an average difference of 0.5% and a maximum difference of 2.5% between the 2D "$x$-$z$" simulation and the 1D analytical solution (Fig. 11). The maximum difference occurs close to the southern tip of the lake, where the thermocline is slightly diffused close to the model boundary.





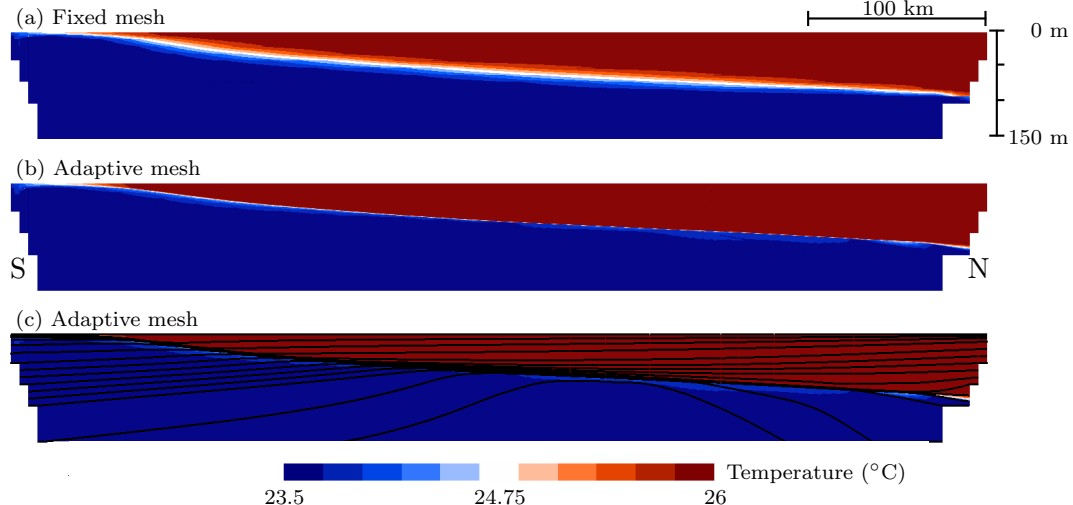

**Figure 12.** Temperature profile using the uniform wind on July 8th 1993, with the 2D "$x$-$z$" model, with a fixed mesh (a) and an adaptive mesh (b,c) having an equal number of elements. (c) also shows the levels position for the moving mesh simulation.

### 3.4 Lake Tanganyika simulation without vertical diffusion

Lake Tanganyika hydrodynamics is simulated using the first model configuration. The model ability to preserve a sharp thermocline is assessed for simulations with and without the vertically adaptive mesh. First, the lake dynamics is simulated with a simple 2D "$x$-$z$" mesh, representing the south-north thermocline position (Fig. 12). Then, it is simulated using a full 3D mesh,

and the thermocline position along the main axis is extracted from the results (Fig. 13). For both 2D "$x$-$z$" and 3D simulations, a sharp thermocline is maintained using an adaptive mesh, while a fixed mesh results in the blurring of the thermocline. Again this confirms that SLIM 3D with vertically adaptive mesh is able to simulate the thermocline dynamics much more accurately than without adaptation.

### 3.5 Lake Tanganyika simulation

Lake Tanganyika dynamics is simulated from December 2000 to April 2004 with the second model configuration, using a simple horizontal mesh of ∼1000 triangles, extruded to form ∼13000 triangular prisms. The horizontal levels are initially located 0, 2 and 5 m, then every 10 m down to 100 m. The next levels are located at 150, 200, 300, 500, 700, 950, 1200, 1400 and 1500 m. The mesh adaptation is driven by the vertical jump in the density field, with $\tau = 1$ h. The background error is set to $f_e = 10^{-5}$ at the surface, and is then defined such that if there is no vertical jump, the mesh stays at its initial position.

Indeed, with a constant background error, the mesh would adapt to reach a situation with the same depth for all the levels at each vertical column. The first 16 months of the simulation are used as a spin-up period, after which the results are analysed.





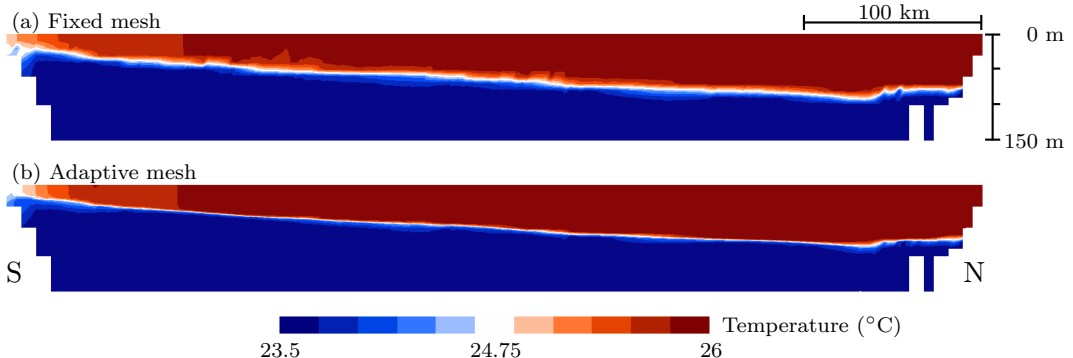

**Figure 13.** Temperature profile using the uniform wind on July 8th 1993, with the 3D model, with a fixed mesh (a) and an adaptive mesh (b) having an equal number of elements. The temperature profile is given along the lake main-axis.

The temporal evolution of the vertical profile of the temperature is analysed at Mpulungu (Fig. 14) and Kigoma (Fig. 15). The model performs well to reproduce most of the observed features of the lake. Outcropping events are observed at Mpulungu in July and August of both 2002 and 2003 (Fig. 14b), such as observed in situ (Fig. 14c). However, the 2003 outcropping lasts much longer in the model than in the data, due to the surface input forcing. The model reproduces the evolution of the 26°C
isotherm during December 2002 - April 2003 period, but fails during the same period of the following year.

At Kigoma, the modelled temperature matches the observations better than at Mpulungu. The 26°C isotherm profile follows the data profile, evolving from 10 to 60 metres deep during the December 2002 - July 2003 period, then from 10 to 50 m from November 2003 to April 2004. This 26°C isotherm outcrops during the dry season strong wind period. A similar observation can be done for the cooler temperatures. The surface temperature is too high during almost the entire year, probably due to the
surface heat flux being biased by the input data.

The thermocline profile along the main axis shows the different regimes of the lake dynamics during the year 2003. At the end of the wet season (March 1st, Fig 16a), the lake is strongly stratified and the thermocline is approximately horizontal. Then, the dry season begins with strong southeasterly winds (May 15th), the surface water flows towards the North (Fig. 17b) and the thermocline is tilted (Fig. 16b). At the end of the dry season, there is a marked outcropping at the southern part of the
lake (Fig. 16c), the wind weakens and the water circulation reverses (Fig. 17c).

Figure 18 shows the 26°C isotherm distribution on (a) March 1st and (b) May 15th, and the 25°C isotherm on August 3rd (c). Figure 18a shows that the temperature is not homogeneous in the direction perpendicular to the main lake axis. East-west temperature gradients may change sign during different periods of the year. This suggests that internal Kelvin waves are travelling along the lake boundary. This temperature gradient switch can occur within a few days in the southern basin of the
lake. Moreover, during the dry season, large outcroppings of water cooler than 26°C (May 15th) and 25°C (August 3rd) are observed in the southern basin.





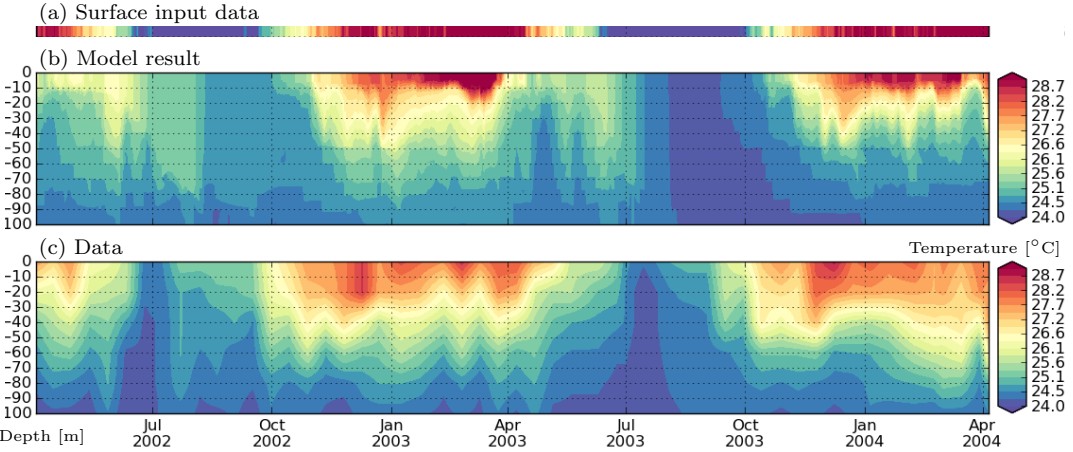

**Figure 14.** Lake water temperature (°C) at Mpulungu as predicted by the model (b) and from in situ observations (c, Descy et al., 2006). Panel (a) shows the surface temperature used to force the temperature flux at the lake surface (Thiery et al., 2015).

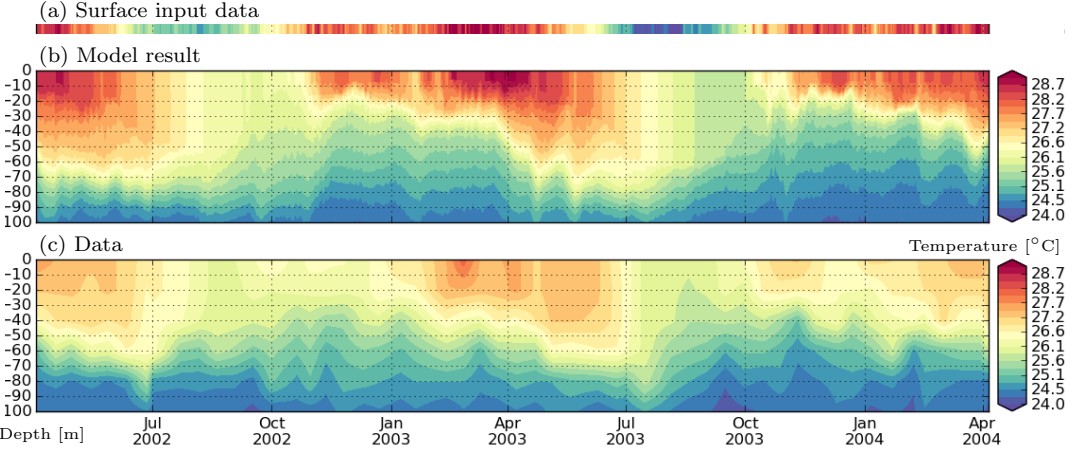

**Figure 15.** Lake water temperature (°C) at Kigoma as predicted by the model (b) and from in situ observations (c, Descy et al., 2006). Panel (a) shows the surface temperature used to force the temperature flux at the lake surface (Thiery et al., 2015).





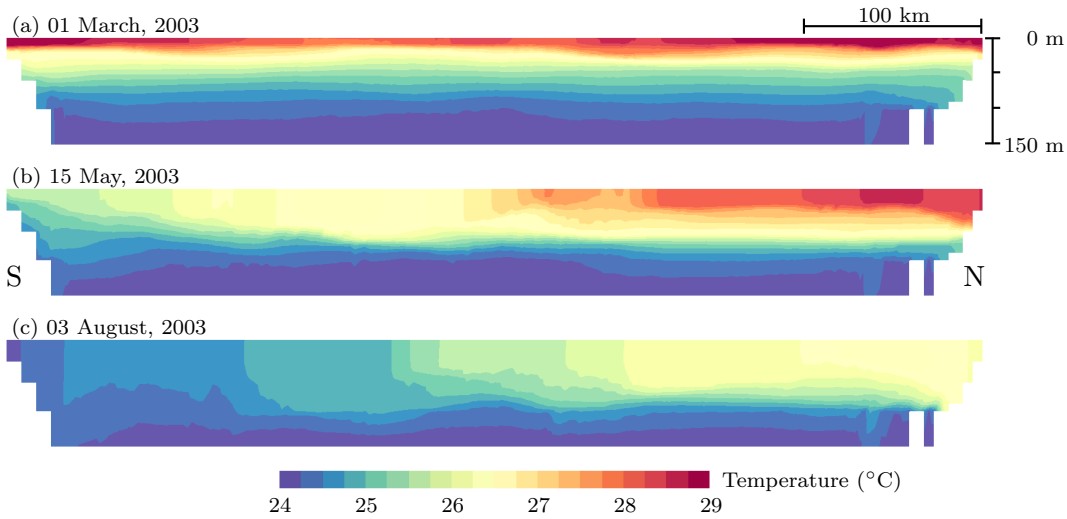

**Figure 16.** South-north temperature transect on March 1st (a), May 15th (b) and August 3rd (c), 2003. Only the 150 upper metres vertical profile are displayed on the figure.

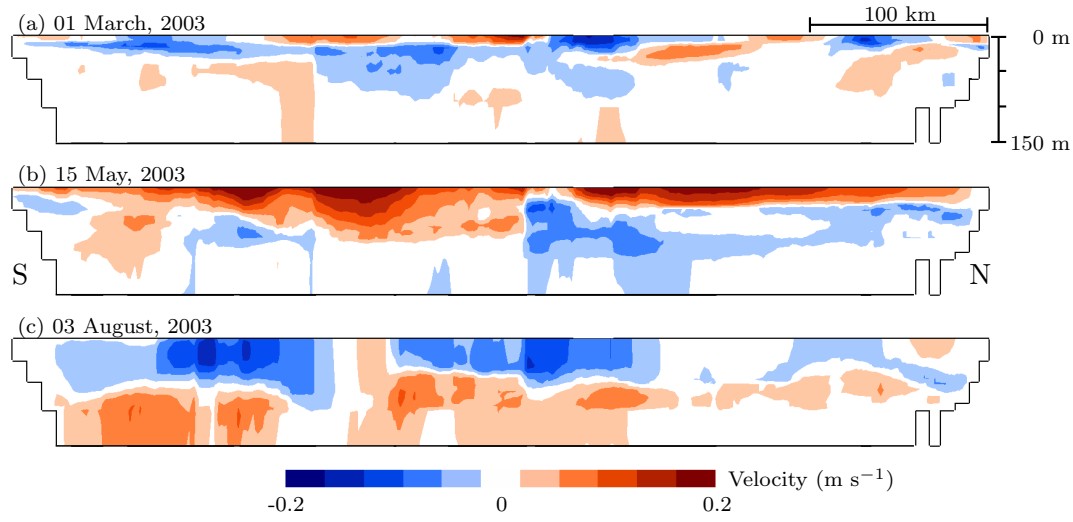

**Figure 17.** South-north velocity transect on March 1st (a), May 15th (b) and August 3rd (c), 2003. Only the 150 upper metres vertical profile are displayed on the figure. Positive value means northward current.



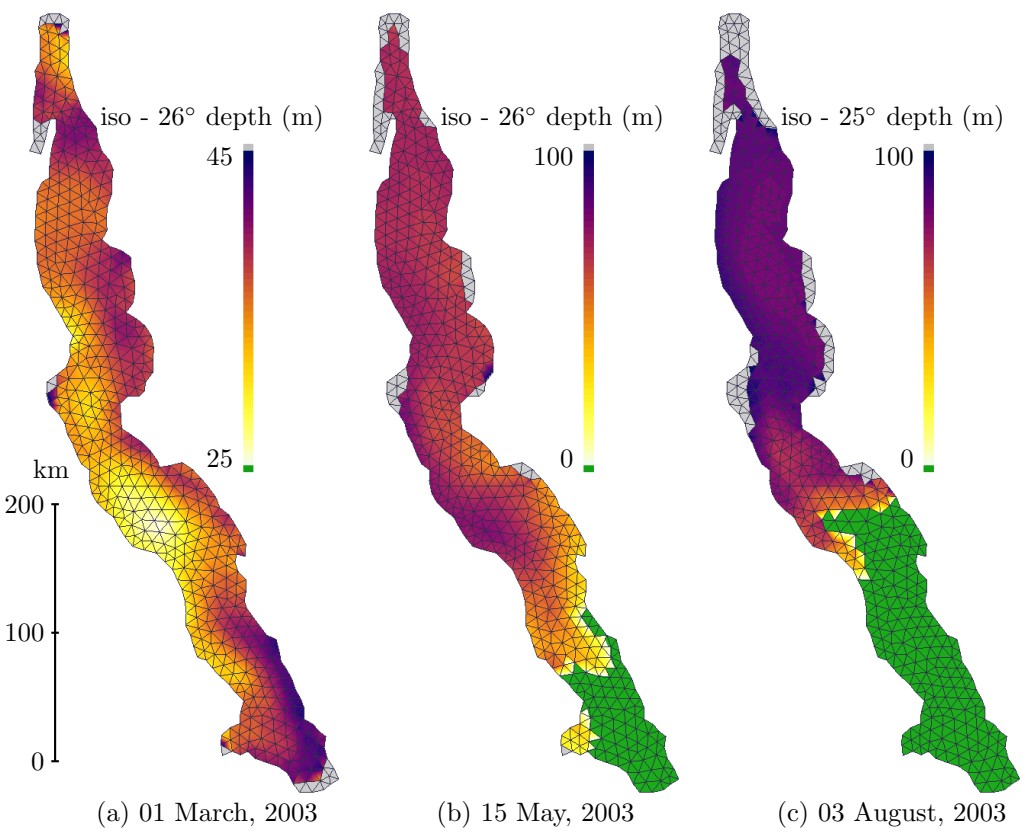

(a) 01 March, 2003        (b) 15 May, 2003        (c) 03 August, 2003

**Figure 18.** Isotherm depths in Lake Tanganyika on March 1st (26°C), May 15th (26°C) and April 3rd (25°C). It is important to note that range of the color bar is not the same through the different maps. Regions in gray represent areas where the bottom water is warmer than the limit temperature. Green regions represent outcropping zones.

# 4 Discussion

## 4.1 Internal seiche and convergence analysis

The internal seiche test case is the typical application for which adaptive coordinates are necessary. The strong discontinuity cannot be preserved using a fixed mesh, which introduces large numerical mixing at the interface (Fig. 8a). The adaptive method introduced in this paper results in a smooth alignment of the levels in the vicinity of the interface (Fig. 8b,c). The method reaches its goal for this benchmark, which is to have one level on each side of the interface with the maximal resolution. It is noteworthy that the numerical mixing at the interface affects the complete dynamics of the lake oscillations. An unresolved interface leads to oscillations with too large a period (Fig. 9). The convergence analysis quantifies the benefit of mesh adaptation, with a gain of the order of 16 in the number of elements and in computation time for a similar error. The





decrease of one order of magnitude in the number of elements is similar to the result obtained by Bernard et al. (2007) with a h-adaptive DG finite element method.

One drawback of the method is that it is necessary to manually set-up the adaptation parameters. In the case of this application, the objective is to maintain the discontinuity at the interface, such that the error is a function of the vertical jumps in the density field. The background error function is a small function just big enough to avoid that a small error in the density field perturbs the mesh smoothness. Eventually, the time relaxation $\tau$ must be small to obtain a fast adaptation. However, if $\tau$ is too small compared to the simulation time step, the smoothness of the moving mesh can be affected.

## 4.2 Steady-state thermocline position

The thermocline slope under a weak constant wind stress test case results in a thermocline slope similar to the analytical solution under the 1D two-layer approximation, with a small deviation close to the southern boundary. This difference is most likely due to the hydrostatic assumption of the model. Indeed, in the overturning circulation, there is an increase of the pressure close to the wall, but this increased pressure is not captured by the model. As a consequence, the boundary layer is not captured by the model and small errors appear in the area irrespective of the horizontal resolution close to the wall. This is not a problem for the 1D model which does not model the overturning circulation within the epilimnion.

For a small, constant wind stress, the analytical 1D solution and the 2D "$x$-$z$" model simulation match very well (Fig. 11). However, for stronger stresses, the epilimnion height decreases such that the two-layer model hypothesis no longer holds. In this situation, the 2D "$x$-$z$" model simulates an outcropping of the hypolimnion layer while the 1D model retains a thin epilimnion layer, as it cannot represent outcropping by construction.

## 4.3 Lake Tanganyika modelling

While the aforementioned test cases are 2D "$x$-$z$" applications, the Lake Tanganyika 3D modelling is more difficult due to the complex coastline, spatial wind patterns and Coriolis effect. Despite this challenge, SLIM 3D realistically represents thermocline oscillations with the adaptive mesh, although a larger relaxation time parameter $\tau$ is necessary to maintain a smooth mesh. In the preliminary simulations, the model runs without vertical diffusivity. In this configuration, a sharp thermocline is maintained under weak wind stress conditions. For a stronger wind that generates outcropping, the thermocline is sligthly diffused near the southern tip of the lake. Indeed, when outcropping occurs, the thermocline is vertical at the border of the outcropping region, and its sharpness depends on the horizontal resolution, which is fixed, independently of the vertical adaptation. During outcropping, the vertical thermocline is thus diffused horizontally. After the wind stress decreases, the thermocline comes back to its original position, but the numerical mixing introduced during the outcropping period cannot be cancelled. However, the thermocline diffusion is much weaker with the adaptive mesh than using a fixed mesh (Figs. 12 and 13).

For the actual Tanganyika runs, vertical diffusivity and viscosity are taken into account. While data scarcity limits a complete validation, the comparison between the model and the available data indicates a good representation of the dynamics of Lake Tanganyika by SLIM 3D. The surface temperature used in the relaxation boundary condition does not match well with the

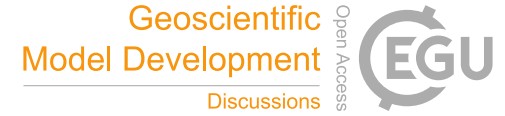



vertical profiles available at Mpulungu and Kigoma, so the modelled vertical profile is biased by construction. But a comparison can still be achieved based on the modelled and observed patterns, such as stratification and thermocline position.

The south-north lake transects of Figure 16 are consistent with the results shown on Figure 8 from Verburg et al. (2011), which modelled the 1996 lake dynamics. In May, which corresponds to the beginning of the strong wind season, the 26.5°C isotherm outcrops at the middle of the lake in both studies, while the 25.5°C isotherm outcrops only at the southern tip of the lake. The agreement between SLIM 3D and Verburg et al. (2011) is also good at the end of this wind season, which corresponds to beginning of August in 2003 and July in 1996.

The southward surface current, going in the opposite direction to the wind suggested by Verburg et al. (2011) is also observed in SLIM results, although this current is most likely due to the weakening of the wind stress and pressure gradient pushing the water mass back to its equilibrium position. This comparison with the results of Verburg et al. (2011) for the year 1996 motivates to investigate further the lake circulation, for a longer period using weather forcings from the COSMO-CLM$^2$ model (Davin and Seneviratne, 2012). This will enable to study the occurrence of events during which there is a reversed overturning circulation.

The presence of internal Kelvin waves in the lake, which was first simulated by means of a 2D reduced gravity model (Naithani and Deleersnijder, 2004) and then demonstrated using scaling arguments supported by laboratory and field investigations (Antenucci, 2005) is also shown with the 3D modelling (Fig. 18). Those waves are more visible during the wet season when the thermocline is almost horizontal.

## 5    Conclusions

A non-uniform vertically adaptive mesh is adjusted for the discontinuous Galerkin (DG) finite element method and implemented into the geophysical and environmental flow model SLIM 3D. The adaptation routine is based on the diffusion of the vertical coordinates, controlled by the vertical jump in the density field.

The adaptation efficiency was tested on simple benchmarks consisting in preserving a sharp interface between two layers of different densities. While the fixed mesh diffuses the interface and produces global errors in the hydrodynamics, the adaptive mesh is able to preserve the interface profile, by aligning thin levels along it. The DG formulation with the mesh adaptation controlled by the vertical jumps preserves the expected field discontinuity with minimal mixing. The necessary manual configuration of the adaptation parameters remains a limitation.

A new formulation for the computation of the mesh vertical velocity, both conservative and consistent, was developed for the adaptive mesh. It is noteworthy that this formulation solves the tracer consistency problem with and without adaptation.

The adaptation was then evaluated by modelling the oscillations of the Lake Tanganyika thermocline. First, a simulation was run without vertical diffusivity and a uniform wind stress, showing the good behaviour of the adaptive mesh. Then, a full simulation of the lake dynamics was performed and compared to time series of vertical temperature profile in the south and the centre of the lake. Overall the outcropping events and the stratification observed on the data are well reproduced by the





model. The remaining differences are partially due to discrepancies between the data used to force the surface heat flux and the validation data.

During the 2-year simulation, the along-axis velocity shows similar patterns to the results from Verburg et al. (2011). To understand better the interactions between the wind velocity, the surface heat flux and the water dynamics, additional simulations

would be necessary. They could be achieved using the full data set available from the COSMO-CLM$^2$ model and the adaptive mesh model SLIM 3D. The model has a strong potential for different applications about the lake hydrodynamics, such as the impact of inter-annual variability and climate change

*Code and data availability.* The SLIM 3D v0.4 code is licensed under GNU GPL v3. It is available through gitlab at https://git.immc.ucl. ac.be/slim/slim. It is archived at Zenodo with the doi 10.5281/zenodo.1002221. The COSMO-CLM$^2$ climate data is also available. It can be

obtained by contacting Wim Thiery (wim.thiery@vub.be).

*Competing interests.* No competing interests are present.

*Acknowledgements.* Computational resources were provided by the Consortium des Équipements de Calcul Intensif (CÉCI), funded by the Belgian Fund for Scientific Research (F.R.S.-FNRS) under Grant No. 2.5020.11. The authors thank the FAO/FINNIDA project GCP/RAF/271/FIN for the measurements used in this study. E. Deleersnijder is an honorary research associate of the F.R.S-FNRS. W. Thiery is supported by an

15 ETH Zurich Fellowship (Fel-45 15-1).





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
