# Peer review of "A fully consistent and conservative vertically adaptive coordinate system for SLIM 3D v0.4 with an application to the thermocline oscillations of Lake Tanganyika"

_Geoscientific Model Development, 2017_

## Referee Comment (RC1) · J. Hill (Referee) · 4 Dec 2017

J. Hill (Referee)

jon.hill@york.ac.uk

Received and published: 4 December 2017

A fully consistent and conservative vertically adaptive coordinate system for SLIM 3D v0.4, a DG finite element hydrodynamic model, with an application to the thermocline oscillations of Lake Tanganyika Philippe Delandmeter et al.

This paper presents a novel method for using vertically adaptive meshes in a DG finite elemennt formulation. The method is implemented in SLIM3D and made avaiable

under a suitabily permissive licence. The method is comprehensively described and then verified on an idealised test case. The new method is then demonstrated on a real-world example to show its capabilities.

The paper is well-written, clear and thorough. I see no major issues with the paper and recommend it be published pending some minor corrections/suggestions.

Minor suggestions:

- Page 2: Line 25. Is it worth making it explicitly clear that hr-adaptivity can add or remove nodes as opposed to r-adaptivity here. There are advantages and disadvantages to both (Piggott et al, 2005). This might then clarify for the reader throughout that the number of nodes in the model remains constant throughout the simulation, which in places is lacking (e.g. figure 10, where the adaptive models look to perform no better than fixed, but of course, their numerical performance is better as the same computational cost). This lack of clarity in the number of nodes being fixed also crops up on line 15 (pg 8) and line 10 (pg 9).

- Page 18, line 10. What was the horizontal resolution?

- Page 21-22. Is it possible to produce a figure or stats on where the mesh resolution was placed alongside these figures? It would be interesting to see the temporal dynamics of the mesh movement.

---

## Author Comment (AC1) · 12 Dec 2017

We would like to thank Prof. Jon Hill for his careful reading and its constructive comments. Please find our replies below.

On behalf of all the authors,

Philippe Delandmeter

This paper presents a novel method for using vertically adaptive meshes in a DG finite element formulation. The method is implemented in SLIM3D and made available under a suitably permissive licence. The method is comprehensively described and then verified on an idealised test case. The new method is then demonstrated on a real-world example to show its capabilities. The paper is well-written, clear and thorough. I see no major issues with the paper and recommend it be published pending some minor corrections/suggestions.

Thank you!

**Minor suggestions:**

Page 2: Line 25. Is it worth making it explicitly clear that hr-adaptivity can add or remove nodes as opposed to r-adaptivity here. There are advantages and disadvantages to both (Piggott et al, 2005). This might then clarify for the reader throughout that the number of nodes in the model remains constant throughout the simulation, which in places is lacking (e.g. figure 10, where the adaptive models look to perform no better than fixed, but of course, their numerical performance is better as the same computational cost). This lack of clarity in the number of nodes being fixed also crops up on line 15 (pg 8) and line 10 (pg 9).

Thank you for highlighting this point. We will follow your suggestion and explain clearly the differences between our r-adaptive method and the hr-adaptive method. As you explained, Figure 10 aims to show that the adaptive method is not more expensive than the non-adaptive method, such that the better result of the adaptive method (in Figure 9) does not require a higher computational cost. This will be written explicitly in the revised manuscript.
**Page 18, line 10. What was the horizontal resolution?**

The horizontal mesh resolution is 10 km. We will add this information in the revised manuscript.

**Page 21-22. Is it possible to produce a figure or stats on where the mesh resolution was placed alongside these figures? It would be interesting to see the temporal dynamics of the mesh movement.**

Yes, the mesh vertical distribution for the Lake Tanganyika simulation will be shown in the revised manuscript, by drawing the levels along the main axis of the lake (similar to Figure 16). It will help the reader to see the mesh dynamics during the simulation and also highlight an important point: in the simpler benchmarks, there is one single large physical discontinuity and the mesh resolution is increased only in this region. But in the realistic simulation of Lake Tanganyika, the thermocline is not that sharp, and the mesh resolution is more regular, although it still follows the thermocline oscillations.

GMDD

---

## Referee Comment (RC2) · Anonymous Referee #2 · 18 Dec 2017

GENERAL REMARKS

The paper presents a new version of 3D hydrodynamic code SLIM 3D v0.4 intended for broad range of marine and limnological applications. The main new feature of the model is an algorithm for vertical grid adaptation to tackle hydrodynamic processes at sharp density gradients. The new model is tested in both idealized scenarios of wind forcing and lake stratification, and realistic simulation of Lake Tanganyika. An

impressive correspondence between the model simulations and analytical solution for the steady-state thermocline tilt is achieved, as well as very good preservation of sharp density gradient at very coarse (6 vertical levels) resolution. Simulations of Lake Tanganyika demonstrate promising capabilities of the model for future studies of the lake circulation and thermal regime. The paper is well structured, the conclusions are clear and enough supported by results presented. Despite the overall high quality of the paper, I see space for improvement, especially in representation of the material and in clarifying some methodological issues.

SPECIFIC COMMENTS

1. The title seems too long. Also consider substituting abbreviation "DG" by the full term.

2. p.4, line 13. When you first refer to "consistency", could you provide definition?

3. There is no general information on the model equation set and boundary conditions

4. Could you provide a clear definition of what is "fixed domain" and what is "moving domain"

5. p.6, line 1. "equations" → "equation"

6. p.6, line 9: please provide explanation to "P1"

7. p.6, line 16: could you explain what is "lateral and horizontal interfaces"?

8. I could see no information on the order of approximation of the model scheme.

9. p.7, line 7. Does the zero vertical velocity at the bottom fits the simulations of Lake Tanganyika with uneven bottom shape? Rather, normal component of velocity should be zero.

10. Not all symbols in equations are explained. I recommend to add a List of symbols.

11. p.8, lines 15-20. I suggest that you provide a 3D picture of the model grid, as this text is somewhat difficult to follow.

12. eq. (15). Is this constant in time or in depth?

13. p.9, line 2 : "adaptivity" → "adaptation"

14. p.9, lines 8-9: the definition is difficult to understand, please consider rephrasing

15. Figure 4 : please explain what do the numbers mean

16. p.10, line 21. At this stage, not clear which "model" is meant

17. p.11, line 5: COSMO-CLM[2], "2" looks like footnote

18. How does modeled stress at Figure 6 compare to measured at Figure 5?

19. Eq. (19) Is there any special reason for parameterizing heat flux by this crude scheme, rather than to apply standard surface flux schemes, based on Monin-Obukhov similarity?

20. Figure 8. What are the black lines? Could you depict the grid levels, at least in the inset?

21. I found no details on which computing system has been used. Was the model parallelized, what number of cores has been utilized?

22. Figure 14. There is larger vertical diffusion of heat in observations, than in the model. What could be the reason?

---

## Author Comment (AC2) · 19 Jan 2018

We would like to thank the anonymous referee for her/his careful reading and constructive comments. Please find our replies below.

On behalf of all the authors,

[Figure]

Philippe Delandmeter

*GENERAL REMARKS The paper presents a new version of 3D hydrodynamic code SLIM 3D v0.4 intended for broad range of marine and limnological applications. The main new feature of the model is an algorithm for vertical grid adaptation to tackle hydrodynamic processes at sharp density gradients. The new model is tested in both idealized scenarios of wind forcing and lake stratification, and realistic simulation of Lake Tanganyika. An impressive correspondence between the model simulations and analytical solution for the steady-state thermocline tilt is achieved, as well as very good preservation of sharp density gradient at very coarse (6 vertical levels) resolution. Simulations of Lake Tanganyika demonstrate promising capabilities of the model for future studies of the lake circulation and thermal regime. The paper is well structured, the conclusions are clear and enough supported by results presented. Despite the overall high quality of the paper, I see space for improvement, especially in representation of the material and in clarifying some methodological issues.*

Thank you!

*SPECIFIC COMMENTS*

*1. The title seems too long. Also consider substituting abbreviation "DG" by the full term.*

Indeed, the title was quite long. To make it shorter, we removed from the title the description of what is SLIM 3D, which is given in the abstract.

*2. p.4, line 13. When you first refer to "consistency", could you provide defini-*

*tion?*

Yes. The term was defined later in the paper. We now define consistency the first time we refer to it.

**3. There is no general information on the model equation set and boundary conditions**

Indeed, this paper does not modify the governing equations that were set in Kärnä et al. (2013). But as you suggested, it is easier for the reader to have the full set of equations in the same manuscript: we added them in a new sub-section at the beginning of the Methods section.

**4. Could you provide a clear definition of what is "fixed domain" and what is "moving domain"**

The governing equations are originally written on a continuously moving domain, since the water volume boundary moves following the free surface. Consequently, the equations cannot be discretised directly, since the mesh does not move continuously but discretely in time. Let's consider a fixed domain, in which the free surface does not move. In such domain the equations can be discretised, but governing equations are not written in this fixed domain. It is why we introduce a mapping to reformulate the original equations from the moving to the fixed domain. This is not new, and it is why we refer also to Formaggia and Nobile (2004).

A better introduction to fixed and moving domains was given in the "Moving mesh and ALE formulation" sub-section.

**5. p.6, line 1. "equations" → "equation"**

Corrected, thank you!

**6. p.6, line 9: please provide explanation to "P1"**

The P1 shape functions are first order polynomial functions. They are bi-linear in case of an extruded triangle mesh and tri-linear for an extruded quad mesh. Since the bi-linear/tri-linear explanation is already given in the next sentence, we simply replaced "P1" by "polynomial" to keep it simple.

**7. p.6, line 16: could you explain what is "lateral and horizontal interfaces"?**

Following your comment 11, a new Figure (Fig. 3) was added. In this figure, we also illustrate what are the lateral and horizontal interfaces.

**8. I could see no information on the order of approximation of the model scheme.**

In SLIM 3D, the equations are approximated using discontinuous piecewise bi-linear/tri-linear functions. This was explained for the moving mesh algorithm, but as you wrote, it was not explained that the entire model follows this approximation. This is now written explicitly in the beginning of the numerical modelling sub-section.

**9. p.7, line 7. Does the zero vertical velocity at the bottom fits the simulations of Lake Tanganyika with uneven bottom shape? Rather, normal component of velocity should be zero.**

Indeed, the impermeability boundary condition requires that the normal component of the water velocity is zero at the bottom. The line you refer says that the vertical mesh velocity ($w_m$) is zero, not the water velocity.

**10. Not all symbols in equations are explained. I recommend to add a List of symbols.**

The different symbols defined in this paper as well as the variables were listed in two tables (Tables 1,2).

**11. p.8, lines 15-20. I suggest that you provide a 3D picture of the model grid, as this text is somewhat difficult to follow.**

A new figure (Fig. 3) was added to illustrate this paragraph using a very simple 3D mesh. Examples of horizontal and vertical interfaces (see comment 7) were also highlighted on the mesh.

**12. eq. (15). Is this constant in time or in depth?**

Thank you for highlighting this confusing term. It is a constant only in depth. This was fixed in the revised manuscript by removing the "constant" term:

$$h_{i+1/2}\, e_{i+1/2} = h_{j+1/2}\, e_{j+1/2}, \quad \forall i, j = 0 \ldots n-1.$$

**13. p.9, line 2 : "adaptivity" → "adaptation"**

Changed, thank you!

**14. p.9, lines 8-9: the definition is difficult to understand, please consider rephrasing**

You are right: the explanation is not straightforward. The upper and lower DG values of a mesh node were illustrated on new Fig. 3 for a better understanding.

[Figure]

**15. Figure 4: please explain what do the numbers mean**

The numbers mean the level depth. It was added in the figure (It is Fig. 5 now) caption.

**16. p.10, line 21. At this stage, not clear which "model" is meant**

Indeed, details about the COSMO-CLM[2] come later. This sentence introduces the section about the two dataset. We rephrased the sentence to remove this unclear "model" word.

**17. p.11, line 5: COSMO-CLM2, "2" looks like footnote**

Indeed, this confusion is understandable. However, the model is called "COSMO-CLM[2]" in the existing literature and we have to keep it this way. Fortunately, the model name is used twice in the same paragraph, so the reader should see that the "2" is part of the model name.

**18. How does modeled stress at Figure 6 compare to measured at Figure 5?**

As you pointed out, one figure shows modelled wind stress while the other shows measured wind stress at one location. We cannot compare them directly, they do not refer to the same calendar year. But beyond those considerations, they both have the same magnitudes during both dry and wet seasons, and we observe on both similar patterns in variability. We didn't compare them directly, but use both of them, the measured (uniform) data for a simple modelling maintaining the sharp interface, and the spatial-varying map for the more realistic simulation.

**19. Eq. (19) Is there any special reason for parameterizing heat flux by this**

*crude scheme, rather than to apply standard surface flux schemes, based on Monin-Obukhov similarity?*

A relaxation flux is not unusual (see e.g. Kamenkovich and Sarachik (Journal of Physical Oceanography, 2004) and Barnier et al. (Journal of Marine Systems, 1995)) and is a simple alternative choice depending on the availability of forcing data. More complex surface fluxes, which require more input data, are currently considered for a better modelling of the surface heat fluxes, but this is not done in the study described here, which focuses on the adaptive moving mesh.

*20. Figure 8. What are the black lines? Could you depict the grid levels, at least in the inset?*

The black lines are precisely the grid levels that you ask to add to the figure. This was highlighted in the caption.

*21. I found no details on which computing system has been used. Was the model parallelized, what number of cores has been utilized?*

SLIM 3D runs on parallel computers. For the Tanganyika simulation, the mesh is rather coarse ($\sim$ 13000 elements) and the simulation was run on 8 CPUs. We also added information about the time step, which was missing.

*22. Figure 14. There is larger vertical diffusion of heat in observations, than in the model. What could be the reason?*

As it is pointed out in the discussion section, the surface temperature used to compute the heat flux (from COSMO-CLM$^2$) does not match accurately with the surface temperature in the two validation locations. Temperature are generally higher in COSMO-CLM$^2$ data than in observations. This induces higher temperature in SLIM 3D

surface results, which increases the stratification. This is most likely a reason of the larger stratification in SLIM 3D. Since stratification is larger, then the water column is more stable and the vertical diffusion is smaller than in observations. But it is noteworthy that this difference in stratification between model and observations isn't very large, and it is relevant to use it to evaluate the quality of the simulation which uses the best available input data. We have improved the paragraph referring to this point in the discussion section.

Please also note the supplement to this comment:
https://www.geosci-model-dev-discuss.net/gmd-2017-221/gmd-2017-221-AC2-supplement.pdf

**Supplement:**

[revised manuscript text omitted]

---

## Author Response (AR1)

**A fully consistent and conservative vertically adaptive coordinate system for SLIM 3D v0.4 with an application to the thermocline oscillations of Lake Tanganyika**

gmd-2017-221

Philippe Delandmeter

January 19, 2018

Dear Editor,

Please find enclosed our revised manuscript "A fully consistent and conservative vertically adaptive coordinate system for SLIM 3D v0.4 with an application to the thermocline oscillations of Lake Tanganyika".

We have amended the manuscript and figures to address the issues raised by both reviewers, and have included a detailed reply to their comments.

You will also find hereinafter a version of the manuscript where the differences with the previous version are highlighted (using latexdiff tool).

Thank you again for considering our manuscript for publication. We are indebted to the two reviewers for their constructive comments. We appreciate your time and look forward to hearing from you.

Yours faithfully,

Philippe Delandmeter
on behalf of all the authors

[revised manuscript text omitted]

---

## Author Response (AR3)

**A fully consistent and conservative vertically adaptive coordinate system for SLIM 3D v0.4 with an application to the thermocline oscillations of Lake Tanganyika**

gmd-2017-221

Philippe Delandmeter

March 4, 2018

Dear Editor,

Indeed, in previous revision I changed "level height" to "level thickness" in the caption of Figure 19, but I forgot to modify the colour bar name. It is now fixed.

Sincerely yours,

Philippe Delandmeter